# RIEMANNIAN OPTIMIZATION FOR HYPERBOLIC PROTOTYPICAL NETWORKS

## ABSTRACT

This paper addresses the utilization of hyperbolic geometry within a Prototype Learning framework. Specifically, we introduce Riemannian optimization for Hyperbolic Prototypical Networks (`RHPN`), a novel approach that leverages Prototype Learning on Riemannian manifolds applied to the Poincaré ball. `RHPN` capitalizes on the efficiency and effectiveness of updating prototypes during training, coupled with a regularization term crucial to boost the performances. We setup an extensive experimentation that shows that `RHPN` is able to outperform the state-of-the-art in Prototype Learning, both in low and high dimensions, extending the impact of hyperbolic spaces to a wider range of scenarios.

## 1 INTRODUCTION

In recent years, there has been a surge of interest among researchers in exploring the embeddings of data onto specific manifolds. This choice can enable the use of metric elements (such as angles and distances) that are particularly suitable to shape the underlying similarity between data, directly in the representation space.

In this context, Prototype Learning (PL) has shown promising results and proved to be a viable alternative to more conventional approaches in various domains, such as image classification Yang et al. (2018); Chen et al. (2019); Mettes et al. (2019), few-shot learning Snell et al. (2017); Dong & Xing (2018), and zero-shot learning Snell et al. (2017); Dong & Xing (2018). The core idea of PL is to build a representation of the target classes within the embedding space (i.e. the *prototypes*), enabling the utilization of metric information to compare new examples with the prototypes and infer the probability of predicting a given class.

Identifying the optimal prototype for a class poses a non-trivial challenge and leveraging prior information on the data can significantly improve the performances. For instance, when the labels are organized hierarchically, changing the geometry used for the embeddings may substantially improve results Landrieu & Garnot (2021); Fonio et al. (2023); Ghadimi Atigh et al. (2021).

When the data exhibit a latent hierarchical structure, it is reasonable to suggest the usage of geometries capable of representing more effectively inter-example distances. Indeed, it is well known that the standard Euclidean geometry does not help in adequately represent trees Linial et al. (1995), while hyperbolic geometries are provably better equipped for dealing with this kind of data Sala et al. (2018); Nickel & Kiela (2018).

In recent years, there has been a surge of interest in the neural networks community about leveraging the unique properties of non-Euclidean geometries. This exploration began with the seminal work by Ganea et al. (2018), and further studied by Ryohei et al. (2021), Van Spengler van Spengler et al. (2023), and Gulcehre Gulcehre et al. (2018).

For what it concerns PL, there have been explorations into non-Euclidean PL within specific data domains Fonio et al. (2023); Hamzaoui et al. (2024); Khrulkov et al. (2020), while tasks such as image classification remain relatively unexplored. The main work tackling the image classification task with an hyperbolic prototypical approach is by Ghadimi Atigh et al. (2021), where the authors prove the effectiveness of hyperbolic manifolds when the embedding space is low-dimensional.

A key characteristic of prototype learning (PL) methods based on neural networks is how prototypes are positioned within the embedding space. Some studies Snell et al. (2017) initialize the prototypes

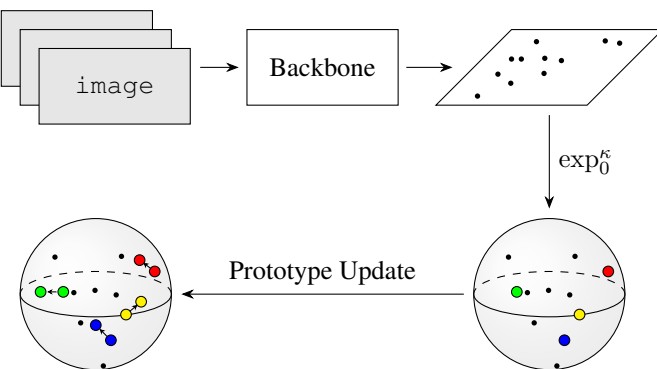

Figure 1: Illustration of Hyperbolic Prototypical Networks on a generic manifold.

at the start of training, allowing the neural network to warp the embedding space so that classification based on those prototypes works effectively. In contrast, other approaches allow the prototypes to move to different regions of the space throughout the learning process Yang et al. (2018).

In this study, we extend this line of research by proposing Riemannian optimization on Hyperbolic Prototypical Networks (RHPN), a prototype learning (PL) framework that exploits hyperbolic geometry and learns prototypes during training by defining them as parameters of the neural network. This approach allows the neural network to optimize the positions of the prototypes (which represent the labels) while considering the data distribution. Furthermore, we incorporate a regularization term that controls the norm of the embeddings, which proves to be a key technique in making hyperbolic representation learning effective *regardless* of the embedding dimension.

In summary, the contributions of this work are threefold:

- we propose a framework for updating prototypes on generic Riemannian manifolds;
- we deeply explore the behavior of the embeddings and the key aspects that make hyperbolic representation learning effective;
- we empirically validate the effectiveness of our method using the Poincaré ball, surpassing existing state-of-the-art methods in PL.

## 2 RELATED WORKS

**Prototype Learning** Prototypical networks are the deep generalization of learning Vector Quantization machines Somervuo & Kohonen (1999) and nearest centroid classifiers Tibshirani et al. (2002).

In most of the approaches, the prototypes are defined as centroids of the representations Snell et al. (2017), positioned a priori Mettes et al. (2019); Fonio et al. (2023); Long et al. (2020) or learnt alongside the training Yang et al. (2018); Landrieu & Garnot (2021). In particular the work from Landrieu & Garnot (2021) introduces the importance of updating the prototypes alongside the training phase and the relevance of adding hierarchical information. On the other hand, Mettes et al. (2019) introduces a non-Euclidean geometry in PL, i.e. the hyperspherical prototypical networks, keeping the prototypes fixed on the hypersphere, maximizing the cosine separation among them. They also highlight the importance of adding hierarchical information in this non-Euclidean context, as investigated also by Fonio et al. (2023). Our approach extends the effort of updating the prototypes on hyperbolic manifolds, defining them as parameters of the network. This approach overcomes the computational issue in calculating the hyperbolic centroid, which is particularly demanding Mettes et al. (2024), and provides better performances than using fixed prototypes as the existing methods do.

**Hyperbolic Representation Learning** Hyperbolic manifolds are claimed to represent hierarchical data with arbitrarily low distortion Sala et al. (2018). For what concerns the hyperbolic mod-

els, five different ones have been defined: Poincaré ball, Lorentz model (Hyperboloid), Poincarè half-plane model, hemisphere model, Beltrami-Klein disk. In the following, we will focus on the Poincaré ball as done by Guo et al. (2022), Khrulkov et al. (2020), van Spengler et al. (2023); Long et al. (2020). While not central to our endeavor, it is worth noting that several works have been based on the Lorentz model Law et al. (2019); Nickel & Kiela (2018), with a comparison of the numerical stability of the Poincaré ball and of the Lorentz model provided in Mishne et al. (2023).

Hyperbolic representations have been also studied in the context of few-shot learning, where some works highlighted the ability of models built on hyperbolic spaces to outperform the state-of-the-art Khrulkov et al. (2020); Hamzaoui et al. (2024). Among these, Khrulkov et al. (2020) gained particular relevance in the recent literature, as it presented the first hyperbolic prototypical framework. However, it is worth noting that this framework is meant for few-shot learning. The adaptation of the proposed method to the image classification task is not considered, nor is it trivial. The use of centroids as prototypes in PL has been proposed by Guerriero et al. (2018), but there is no trace of adaptation to a hyperbolic setting. A few works have tackled the problem of exploiting hyperbolic geometries for image classification. Ghadimi Atigh et al. (2021) pioneered this approach, while Yue et al. (2024) explored the impact of changing the temperature parameter in a contrastive loss when exploring a hyperbolic space. Ghadimi Atigh et al. (2021) introduces a method with fixed ideal prototypes positioned on the boundary of the Poincaré ball, which is conceptually at an infinite distance from the center of the hyperbole. To overcome the problems derived from placing the prototypes in this way, the authors introduce the usage of the Busemann distance to make it possible to compare a point on the manifold and the prototypes.

In our work we place the prototypes within the Poincaré ball (i.e., not on the boundary), but allow them to be updated to better capture the structure of the data distribution.

## 3 BACKGROUND

**Definition 1.** *A **manifold** $\mathcal{M}$ of dimension $n$ is a topological space such that each point's neighborhood can be locally approximated by the Euclidean space $\mathbb{R}^n$.*

In this paper we are considering the **Poincaré ball**, i.e.:

$$\mathbb{B}_\kappa^n = \{x \in \mathbb{R}^n : \kappa\|x\|^2 < 1\}$$

**Definition 2.** *Given a point $x \in \mathcal{M}$, the **tangent space** $\mathcal{T}_x\mathcal{M}$ of $\mathcal{M}$ at $x$ is the $n$-dimensional vector-space, omeomorphic to $\mathbb{R}^n$, built as the first order approximation of $\mathcal{M}$ around $x$.*

**Definition 3.** *The **Riemannian metric** is the metric tensor that gives a local notion of angle, length of curves, surface and volume. For a manifold $\mathcal{M}$, the Riemannian metric $g_x$ is a smooth collection of inner products on the associated tangent space: $g_x : \mathcal{T}_x\mathcal{M} \times \mathcal{T}_x\mathcal{M} \to \mathbb{R}$. A **Riemannian manifold** is defined as a manifold equipped with a Riemannian metric $g$, and is written $(\mathcal{M}, g)$.*

The Riemannian metric of $\mathbb{B}^n$ is given by:

$$g_x^B = \lambda_x^\kappa g^E, \tag{1}$$

with:

$$\lambda_x^\kappa = \frac{1}{1 - \kappa\|x\|_2^2}, \tag{2}$$

where $x \in \mathbb{B}^n$, $\kappa$ is the curvature of the hyperbolic manifold, and $g^E = I_n$ is the Euclidean metric (the identity matrix).

**Definition 4.** *A **geodesics** $\gamma$ is the shortest path between two points on the manifold. It can be seen as the generalization of the straight line in Euclidean spaces. Given $x, y \in \mathcal{M}$ the **distance** $d(x, y)$ is defined by measuring the length of the geodesic segment connecting the two points.*

In the case of the Poincaré ball, we have:

$$d_\kappa(x, y) = \frac{2}{\sqrt{\kappa}} \tanh^{-1}(\sqrt{\kappa}\|(-x) \oplus_\kappa y\|_2) \tag{3}$$

**Definition 5.** *Given a point $x \in \mathcal{M}$ and a vector $v \in \mathcal{T}_x\mathcal{M}$, the **exponential map** projects $v$ to the manifold $\mathcal{M}$, $\exp_x^\kappa(v) : \mathcal{T}_x\mathcal{M} \to \mathcal{M}$. The projection is obtained by moving the point along the geodesic $\gamma : [0,1] \to \mathcal{M}$ uniquely defined by $\gamma(0) = x$ and $\gamma'(0) = v$. The projection is defined to be $\exp_x^\kappa(v) = \gamma(1)$. The precise definition of the exponential map depends on the manifold; its inverse function is called the **logarithmic map**, $\log_x^\kappa(\cdot)$.*

If $\mathcal{M} = \mathbb{B}_\kappa^n$ and $x = 0$:

$$\exp_0^\kappa(v) = \frac{1}{\sqrt{\kappa}} \tanh(\sqrt{\kappa}\|v\|/2)\frac{v}{\|v\|}. \qquad (4)$$

As a measure to calculate the hyperbolicity of the embeddings, the $\delta$-hyperbolicity has been generally accepted by the community. It is a well-established mathematical tool that relies on calculating the curvature of a space, taking into account triplets and quadruplets of points. In the pioneering work by Khrulkov et al. (2020), the authors used $\delta$-hyperbolicity as a score to justify the embedding of images onto an hyperbolic manifold. A lower $\delta$-hyperbolicity indicates a stronger hyperbolicity. The following definition is taken from Khrulkov et al. (2020).

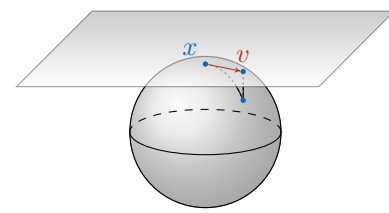

Figure 2: Illustration of the exponential map (Eq. 4) applied onto a generic manifold $\mathcal{M}$ (shown as a sphere in the picture).

**Definition 6.** *Let $X$ be a metric space endowed with the distance function $d$. The **Gromov Product** for $x, y, z \in X$ is defined as:*

$$(y, z)_x = \frac{1}{2}(d(x, y) + d(x, z) - d(y, z)).$$

**Definition 7.** *The $\delta$-hyperbolicity of the metric space $X$ is the minimal $\delta$ value such that for any $x, y, z, w \in X$:*

$$(x, z)_w \geq \min((x, y)_w, (y, z)_w) - \delta.$$

## 4 METHOD

Yang et al. (2018) and Landrieu & Garnot (2021) introduced the idea of extending deep networks to learn prototypes by embedding the prototype representations as network parameters. In this section we introduce RHPN, which extends this idea to work with non-Euclidean geometries and, in Section 5.2, we show how crucial aspects of hyperbolic embeddings impact on the performances.

Our methodology involves extracting the output from a backbone network, such as a ResNet18, and projecting it onto the Poincaré ball. Subsequently, distances from class prototypes are computed. These distances are interpreted as a probability distribution using softmax activation, which is then employed in a cross-entropy loss function for learning purposes.

We assume to be given a dataset $X = \{(x_i, y_i)\}_{i=1}^N$, with $x_i$ taking values in a sample space $\mathcal{X}$, $y_i \in \mathcal{C} = \{1 \ldots C\}$, and $|X| = N$. A backbone network $f(\cdot, \theta) : \mathcal{X} \to \mathbb{R}^d$ is augmented with parameters $\Pi = \{\pi_j, j \in \mathcal{C}\}$ representing the prototypes, with $\pi_j \in \mathbb{B}_\kappa^d$ (here and in the following, we will assume $\kappa = 1$). Prototypes $\pi_j$ are initialized by sampling randomly from $[-0.1, 0.1]$ and then projecting the sampled points onto the Poincaré ball via the exponential map. We train a RHPN model, by solving:

$$\arg\min_{\theta, \Pi} \mathcal{L}(\theta, \Pi; \gamma),$$

i.e., finding the parameters $\theta$ and $\Pi$ minimizing over the training set the distance based cross-entropy loss Yang et al. (2018):

$$\mathcal{L}(\theta, \Pi; \gamma) = \frac{1}{N} \sum_{(x_i, y_i) \in X} -\log \frac{e^{-\gamma d(z_i, \pi_{y_i})}}{\sum_{j \in \mathcal{C}} e^{-\gamma d(z_i, \pi_j)}}, \qquad (5)$$

where $z_i = \exp_0(f(x_i)) \in \mathbb{B}_1^d$, $f$ is the backbone network, $\gamma$ is the *temperature* parameter Yang et al. (2018), and $d(\cdot, \cdot) \equiv d_1(\cdot, \cdot)$ is the distance defined in equation 3 with $\kappa = 1$.

It is important to notice that prototypes within this context possess a dual nature: they function as parameters of the architecture, while existing as entities within the embeddings space. Computing the gradient involves operating within the parameter space, but it is instead crucial to maneuver the prototypes within the embedding space (i.e. the Poincaré ball).

For this reason we update the prototypes using the Riemannian SGD Bonnabel (2013) update rule:

$$\pi \leftarrow \exp_\pi(-\mu \cdot \nabla_\pi^R \mathcal{L}), \tag{6}$$

where $\nabla_x^R = \nabla_x/(\lambda_x^\kappa)^2$ is the Riemannian gradient, $\lambda_x^\kappa$ is defined in equation 2, and $\nabla_x$ is the standard euclidean gradient. The intuition about how the update rule operates is that the exponential map folds the gradient vector on the tangent space onto the Poincaré ball (see Figure 3).

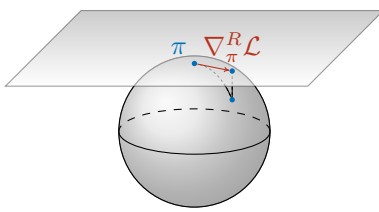

As shown by Guo et al. (2022), embeddings with norm close to the boundary can bring to a vanishing gradient problem. To overcome this problem, we clip the features to be at most 1 before applying the exponential map, which corresponds to clipping the norm of the embeddings in the Poincaré ball to 0.76.

To further pressure the embeddings to sway afar from the boundary, we adopt the regularization term proposed in (Ghadimi Atigh et al., 2021):

$$\mathcal{L}_{reg}(z) = -\lambda \cdot d \log(1 - \|z\|^2), \tag{7}$$

Figure 3: Illustration of the Riemannian SGD update rule.

where $d$ is the embedding dimension. The parameter $\lambda$, referred to as the *slope*, regulates the extent to which the embeddings are drawn towards the center. As shown in Section 5, our method exhibits significant sensitivity to variations in the slope parameter.

## 5 EXPERIMENTS

We tested `RHPN` over 4 datasets and compared its results against 4 methods that use PL: *HPS* exploits an hyperspherical geometry Mettes et al. (2019) with fixed prototypes; *CHPS* exploits an hyperspherical geometry and optimizes a similarity-based cross-entropy loss Fonio et al. (2023), with fixed prototypes; *ECL* exploits a Euclidean geometry and optimizes a distance-based cross-entropy loss, updating the prototypes Landrieu & Garnot (2021). *HBL* exploits an hyperbolic geometry Ghadimi Atigh et al. (2021) with fixed prototypes. As a further baseline, *XE* exploits a traditional training minimizing a cross-entropy loss. To the best of our knowledge, `RHPN` is the first non-Euclidean method where prototypes are not fixed. We refer to `RHPN` for our proposed methodology *without* regularization ($\lambda = 0$) and `RHPN*` for the best $\lambda$ value. In our experiments, for `RHPN` we used clip value equal to 1, curvature of the space fixed to 1, temperature in Eq.5 $\gamma = 10$. For the slope parameter, after a hyperparameter optimization we have kept 0.1 for the first dimension of each dataset and 0.01 from the second dimension on. For what concerns the backbone, each method trained a ResNet18 from scratch. Further details about the experiments are available in appendix A.

For what concerns the datasets, we have chosen benchmark datasets in Computer Vision for Fine-Grained image classification.

**Cifar-100** (Krizhevsky, 2009) 100 classes, $50000/10000$ examples (train/test);

**CUB** (Wah et al., 2011) 200 classes, $5994/5794$ examples (train/test);

**Aircraft** (Maji et al., 2013) 100 classes, $6667/3333$ examples (train/test);

**Cars** (Krause et al., 2013) 196 classes, $8144/8041$ examples (train/test).

We have reproduced the competitors in their setting. More details about the training procedure and the hyperparameter-optimization are in the Appendix A. For complete reproducibility, we release the code, available at `https://anonymous.4open.science/r/RHPN-ICLR25-D029`.

### 5.1 RESULTS

We present the results of our experiments in Table 1, Table 2, Table 3 and Table 4. They show the average test accuracy over 3 runs (with their standard deviation) for each method and each dataset.

Table 1: Percentage of test accuracy on CUB for our proposed method and the competing approaches. The best results among all the methods are in bold, while the second best method is underlined.

| Method | Embedding dimension | | | |
|---|---|---|---|---|
| | **16** | **64** | **128** | **200** |
| XE | - | - | - | $48.17_{\pm 0.59}$ |
| HPS | $14.33_{\pm 0.20}$ | $39.53_{\pm 0.71}$ | $43.91_{\pm 0.70}$ | $46.76_{\pm 0.46}$ |
| CHPS | $27.00_{\pm 0.97}$ | $40.19_{\pm 0.87}$ | $45.19_{\pm 0.69}$ | $46.37_{\pm 0.33}$ |
| ECL | $\underline{36.03}_{\pm 0.51}$ | $\underline{43.64}_{\pm 1.49}$ | $\underline{47.84}_{\pm 0.76}$ | $\underline{50.16}_{\pm 0.53}$ |
| HBL | $28.99_{\pm 0.87}$ | $27.28_{\pm 1.39}$ | $45.34_{\pm 1.00}$ | $44.10_{\pm 1.58}$ |
| RHPN | $39.96_{\pm 0.84}$ | $45.96_{\pm 1.08}$ | $47.16_{\pm 0.48}$ | $46.97_{\pm 0.72}$ |
| RHPN* | $\mathbf{46.28}_{\pm 0.71}$ | $\mathbf{50.44}_{\pm 0.91}$ | $\mathbf{52.68}_{\pm 0.18}$ | $\mathbf{53.19}_{\pm 0.49}$ |
| Improvement | 10.25 | 6.8 | 4.84 | 3.03 |

Table 2: Percentage of test accuracy on Cars for our proposed method and the competing approaches. The best results among all the methods are in bold, while the second best method is underlined.

| Method | Embedding dimension | | | |
|---|---|---|---|---|
| | **16** | **64** | **128** | **196** |
| XE | - | - | - | $\underline{73.21}_{\pm 0.53}$ |
| HPS | $7.18_{\pm 2.09}$ | $42.82_{\pm 2.38}$ | $57.09_{\pm 1.45}$ | $60.94_{\pm 0.96}$ |
| CHPS | $25.23_{\pm 8.14}$ | $55.77_{\pm 1.27}$ | $63.58_{\pm 0.49}$ | $66.63_{\pm 0.64}$ |
| ECL | $\underline{60.66}_{\pm 1.58}$ | $\underline{65.94}_{\pm 0.62}$ | $\underline{69.13}_{\pm 0.10}$ | $70.69_{\pm 1.22}$ |
| HBL | $32.68_{\pm 0.73}$ | $38.63_{\pm 1.67}$ | $58.28_{\pm 2.95}$ | $46.48_{\pm 2.77}$ |
| RHPN | $65.22_{\pm 1.22}$ | $69.15_{\pm 0.74}$ | $70.54_{\pm 0.59}$ | $71.21_{\pm 0.83}$ |
| RHPN* | $\mathbf{69.56}_{\pm 0.81}$ | $\mathbf{72.45}_{\pm 0.69}$ | $\mathbf{75.86}_{\pm 1.12}$ | $\mathbf{77.68}_{\pm 0.38}$ |
| Improvement | 8.9 | 6.51 | 6.73 | 4.47 |

For each dataset, we report experiments done varying the embedding dimension, we experimented with embedding dimensions 16, 64, 128 for CUB and Cars and with embedding dimensions 8, 32, 64 for Aircraft and Cifar100. For each of them, we also report the results where the embedding dimension is equal to the number of classes. It is worth noting that previous studies employing hyperbolic geometries in image classification Ghadimi Atigh et al. (2021), emphasized the good performances of hyperbolic spaces only in low dimensional settings, while we experiment using a wider range of embedding dimensions.

In general, lower-dimensional embeddings negatively impact performances, as the same information must be encoded in a less expressive space. For datasets with a large number of classes, this issue is exacerbated, as they require more expressive feature spaces to effectively differentiate among the increased number of categories.

Table 1 and Table 2 reports results of ours and competing methods over the CUB and Cars datasets. We note that these datasets are the ones having the higher number of classes and our method seems to be particularly effective. Specifically, the accuracy gap between RHPN and the second-best competitor (typically *ECL*) is more pronounced in CUB and Cars datasets compared to Aircraft and Cifar100 (see also Tables 3 and 4).

However, the effectiveness of RHPN is not limited to dataset with a high number of classes. Our experiments clearly demonstrate that RHPN consistently outperforms state-of-the-art methods *regardless* of the embedding dimension, on every dataset, except for dimension 8 in Cifar100.

Our experiments show that a performance loss when the embedding dimension is reduced is indeed to be expected. Figure 6 makes this observation evident by reporting the loss in performances that each method suffers when the embedding dimension is reduced from the maximal allowed. Interestingly, RHPN shows to be very robust in these regards. Particularly on datasets with a large number

Table 3: Percentage of test accuracy on Aircraft for our proposed method and the competing approaches. The best results among all the methods are in bold, while the second best method is underlined.

| Method | Embedding dimension | | | |
|---|---|---|---|---|
| | **8** | **32** | **64** | **100** |
| XE | - | - | - | $76.65 _{\pm 0.36}$ |
| HPS | $29.76 _{\pm 5.94}$ | $66.88 _{\pm 2.08}$ | $69.79 _{\pm 0.53}$ | $73.06 _{\pm 0.63}$ |
| CHPS | $59.95 _{\pm 0.47}$ | $72.75 _{\pm 0.74}$ | $74.56 _{\pm 0.52}$ | $75.87 _{\pm 0.80}$ |
| ECL | $\underline{75.27} _{\pm 0.74}$ | $\underline{77.81} _{\pm 0.34}$ | $\underline{78.15} _{\pm 0.45}$ | $\underline{78.06} _{\pm 0.37}$ |
| HBL | $57.18 _{\pm 2.55}$ | $70.75 _{\pm 0.41}$ | $61.67 _{\pm 0.34}$ | $63.89 _{\pm 0.81}$ |
| RHPN | $76.68 _{\pm 0.06}$ | $78.13 _{\pm 0.44}$ | $77.65 _{\pm 0.41}$ | $78.77 _{\pm 0.31}$ |
| RHPN* | $\mathbf{78.11} _{\pm 0.95}$ | $\mathbf{80.17} _{\pm 0.68}$ | $\mathbf{80.83} _{\pm 0.38}$ | $\mathbf{81.59} _{\pm 0.45}$ |
| Improvement | 2.84 | 2.36 | 2.68 | 3.53 |

Table 4: Percentage of test accuracy on Cifar100 for our proposed method and the competing approaches. The best results among all the methods are in bold, while the second best method is underlined.

| Method | Embedding dimension | | | |
|---|---|---|---|---|
| | **8** | **32** | **64** | **100** |
| XE | - | - | - | $\underline{75.63} _{\pm 0.26}$ |
| HPS | $54.37 _{\pm 1.32}$ | $67.38 _{\pm 0.67}$ | $68.96 _{\pm 0.14}$ | $69.17 _{\pm 0.25}$ |
| CHPS | $70.18 _{\pm 0.48}$ | $73.71 _{\pm 0.15}$ | $74.16 _{\pm 0.34}$ | $74.16 _{\pm 0.34}$ |
| ECL | $\mathbf{74.09} _{\pm 0.43}$ | $\underline{74.31} _{\pm 0.19}$ | $\underline{74.48} _{\pm 0.19}$ | $74.28 _{\pm 0.24}$ |
| HBL | $70.71 _{\pm 0.30}$ | $72.03 _{\pm 0.28}$ | $72.23 _{\pm 0.36}$ | $70.32 _{\pm 1.15}$ |
| RHPN | $72.29 _{\pm 0.18}$ | $74.73 _{\pm 0.11}$ | $74.95 _{\pm 0.20}$ | $75.16 _{\pm 0.21}$ |
| RHPN* | $\underline{73.26} _{\pm 0.40}$ | $\mathbf{75.53} _{\pm 0.05}$ | $\mathbf{76.31} _{\pm 0.06}$ | $\mathbf{76.63} _{\pm 0.28}$ |
| Improvement | -0.83 | 1.22 | 1.83 | 1.00 |

of classes, RHPN achieves the smallest performance degradation as the embedding dimension is reduced (see Figure 6d and Figure 6b). On the other hand, as shown in Figure 6c and Figure 6a, when the number of classes is small, the method suffering the lowest degradation in performances is the euclidean approach implemented in ECL. We further notice that, as expected, angle-based methods (i.e. *HPS* and *CHPS*) suffer dramatically from low-dimensional spaces. Last, somewhat unexpectedly, there are a few cases where the *HBL* method seem to benefit from lowering the embedding dimension.

Overall, RHPN shows to be more robust to dimensionality changes w.r.t. the competitors, in that its performances are seldom and/or only mildly affected by the change.

However, a key aspect that impacts on the performances is the slope ($\lambda$) parameter. In all results presented so far, RHPN is always dominated by RHPN*, which highlights the beneficial role of regularising the norms of the embeddings. Figure 4a and Figure 4b show how performance vary as the $\lambda$ parameter changes. In all our experiments, $\lambda = 0.1$ proved to be the best one when the embedding dimension is low, while $\lambda = 0.01$ works better for high embedding dimensions. In Figure 4c and Figure 4d we show how the average norm of the embeddings behaves during learning in different settings for the $\lambda$ parameter.

## 5.2 $\delta$-Hyperbolicity

Khrulkov et al. (2020) used the $\delta$-hyperbolicity of the embeddings extracted from pre-trained visual models to justify the use of hyperbolic spaces in Computer Vision. We believe it is interesting to investigate this metric properly so to have an overview of the behavior of the embeddings built by RHPN and by the competitors so to better provide a justification to using hyperbolic manifolds. In the

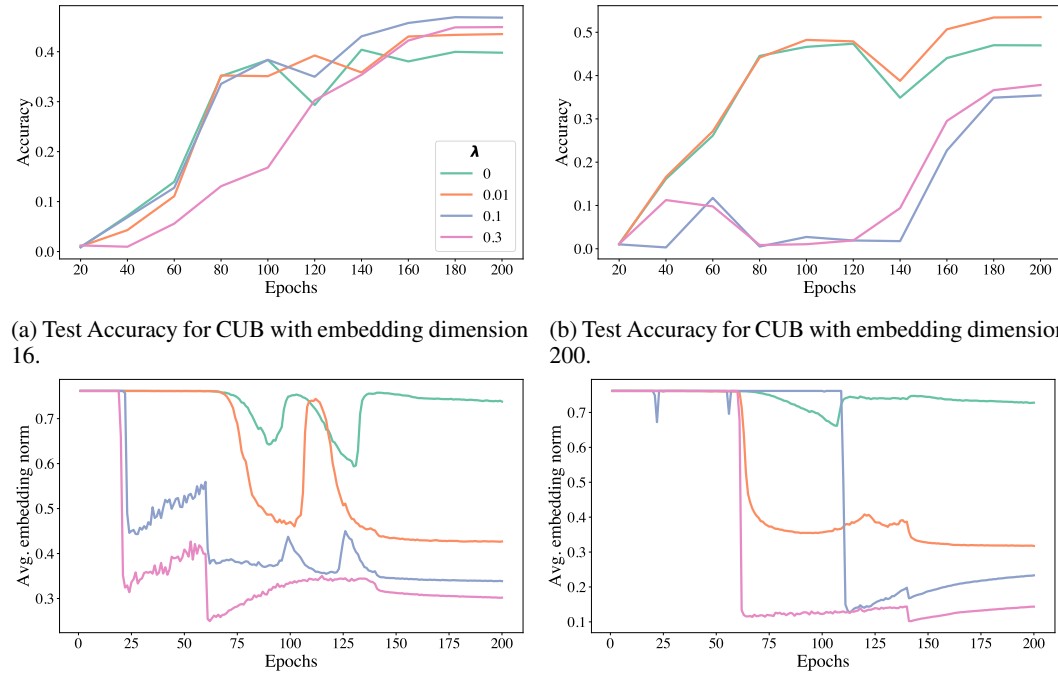

(a) Test Accuracy for CUB with embedding dimension 16.

(b) Test Accuracy for CUB with embedding dimension 200.

(c) Average embedding norm for CUB with embedding dimension 16.

(d) Average embedding norm for CUB with embedding dimension 200.

Figure 4: Impact of the slope parameter on the average embedding norm and the performances.

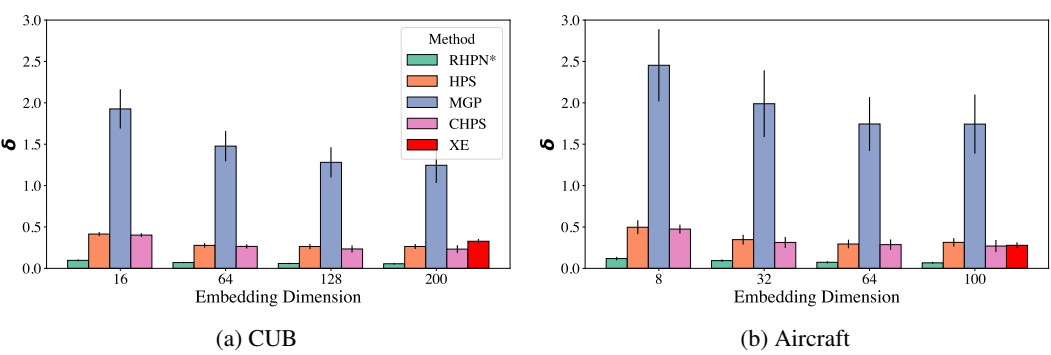

(a) CUB

(b) Aircraft

Figure 5: Illustration of the $\delta$-hyperbolicity according to the different geometries used in the embedding space.

case of RHPN, we compute the $\delta$-hyperbolicity on the embeddings built by the backbone network, i.e., *before* projecting them on the Poincaré ball. For the hyperspherical methods (HPS and CHPS), we use the normalized embeddings as devised in the respective methods. In all other cases, as no projection is needed, we compute the measure directly on the output of the Neural Network. The $\delta$-hyperbolicity is measured for each batch of the test dataset, and then averaged.

From our experiments it is clear that the geometry used during training affects the hyperbolicity of the embeddings. As we can see from Figure 5, the embeddings learnt by our method show a very low $\delta$ when compared to the other methods (i.e., they are more likely to live on a hyperbolic manifold than the embeddings built by the competitors). Since the initial dataset is the same for all methods, this observation suggests that employing a hyperbolic geometry during learning, guides the embeddings to conform to it, hopefully better aligning them to the intrinsic structure of the data.

We also observe that the *slope* hyper-parameter in the loss can significantly affect the hyperbolicity of the embeddings. It is worth mentioning that embeddings near the boundary of the Poincaré ball

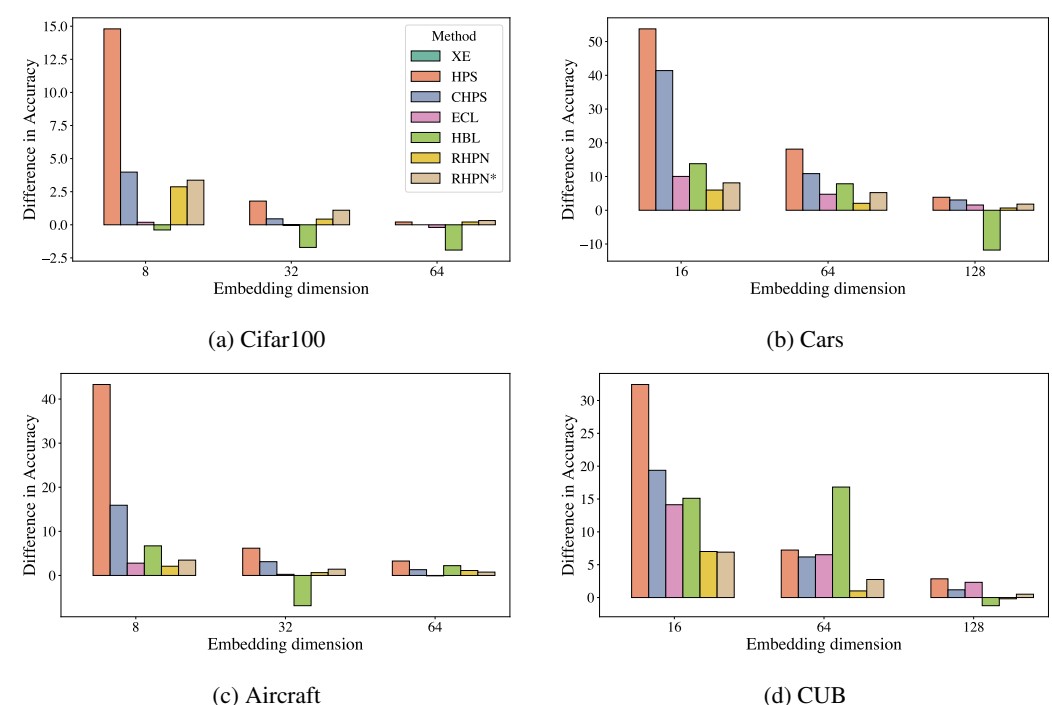

(a) Cifar100

(b) Cars

(c) Aircraft

(d) CUB

Figure 6: Difference in accuracy between the highest embedding dimension tested (i.e. the number of classes) and the corresponding dimension on the x-axis.

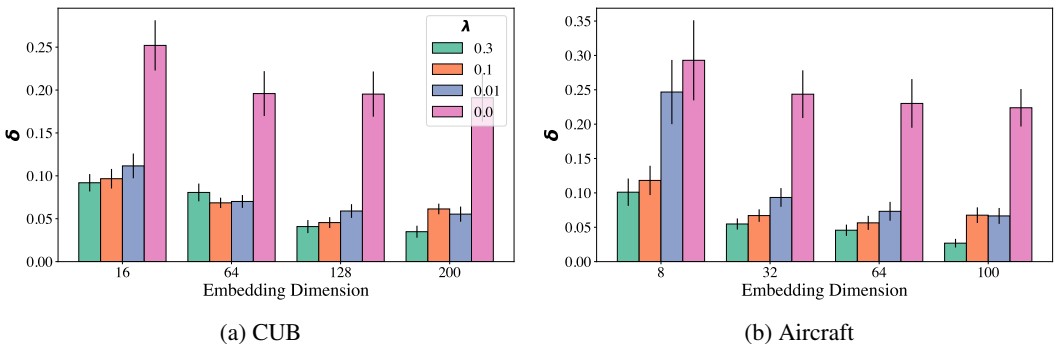

(a) CUB

(b) Aircraft

Figure 7: Illustration of the $\delta$-hyperbolicity according to the different slopes using RHPN.

are supposed to show an hyperspherical behavior, rather than a hyperbolic one. However, in our experiments this is not observed. In fact, when using $\lambda = 0$, RHPN is able to learn embeddings that are close to the boundary and still have $\delta$ that are visibly smaller than those obtained by hyperspherical methods (see Figures 7 and 5). In addition, as expected, embeddings learnt with larger slopes (i.e. closer to the center) appear to have higher hyperbolicity (lower $\delta$). It is important to stress that we are not implying that higher hyperbolicities are necessarily related to better performances. Indeed, Figure 8 shows that very high slope values (leading to higher hyperbolicities) can bring the embeddings to collapse towards the center, hindering learning performances. This is particularly harmful for high-dimensional spaces as shown in Figure 8b and Figure 8d. It naturally follows that the $\delta$ parameter needs to be carefully chosen to guarantee the correct "amount" of hyperbolicity to the learnt embeddings.

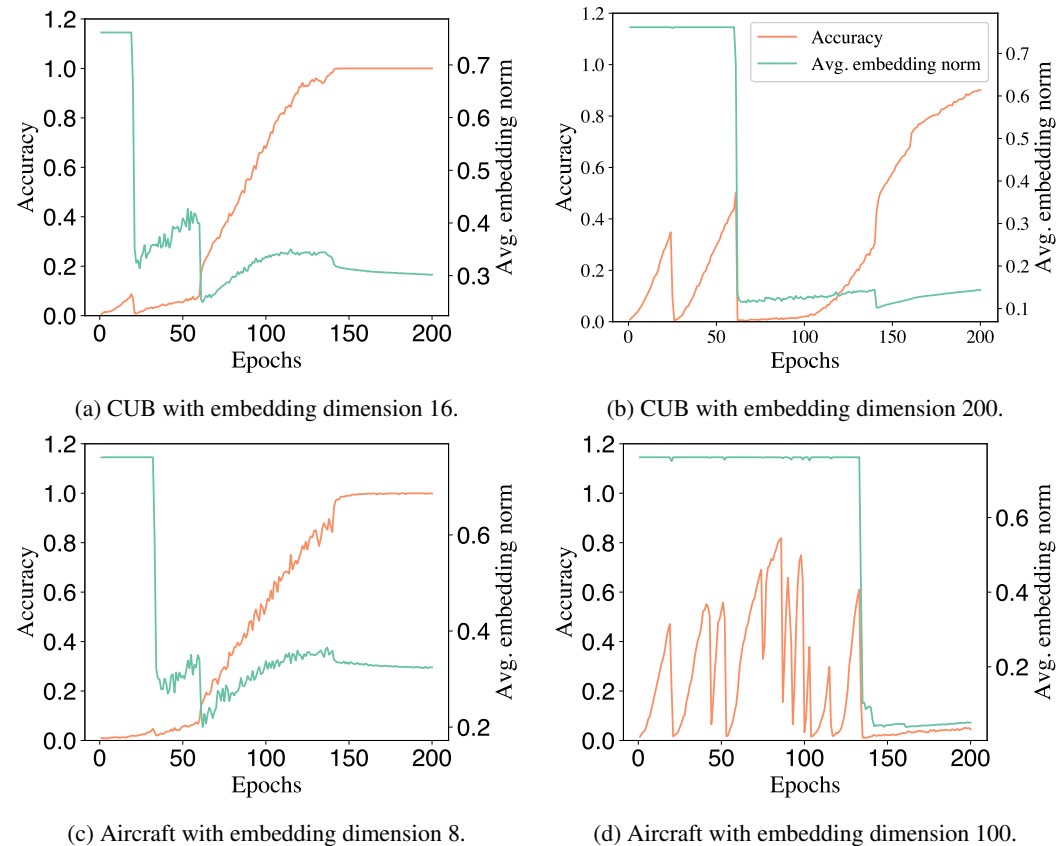

(a) CUB with embedding dimension 16.

(b) CUB with embedding dimension 200.

(c) Aircraft with embedding dimension 8.

(d) Aircraft with embedding dimension 100.

Figure 8: Training performances and average embedding norm in presence of a slope $\lambda = 0.3$.

## 6 CONCLUSIONS

In conclusion, this study has introduced a methodology for dynamically updating the prototypes during the training process within the context of hyperbolic representation learning. `RHPN` extends the effectiveness of hyperbolic manifolds to any embedding dimension, leveraging the importance of controlling the norm of the embeddings. We validate our findings through a wide range of experiments.

For future work, we plan to leverage background knowledge to improve the initialization and positioning of prototypes during training. Additionally, we intend to explore other geometries, as `RHPN` is flexible and not restricted to the Poincaré model.

### REPRODUCIBILITY STATEMENT

To reproduce completely our experiments we provide the code in Section 5. The details of the experimental setting, as well as the hardware capacities, can be found both in Section 5 and in the Appendix A.

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

## A APPENDIX

### A.1 EXPERIMENTAL DETAILS

RHPN was trained for 200 epochs with SGD, using a learning rate of 0.1, weight decay 0.001, momentum 0.9, and linear learning rate scheduler at epochs 60, 120 and 160. The backbone used was a ResNet18. As discussed in section4, to update the prototypes we used RSGD Bonnabel (2013) with learning rate 0.001 and momentum 0.9. The batch size was set to 128 for Aircraft and Cifar100Krizhevsky et al. (2009) and to 64 for the other datasets. The embedding dimension was set to 8 for Cifar10, 64 for Cifar100 and AircraftMaji et al. (2013), and 128 for CubWah et al. (2011) and CarsKrause et al. (2015). The choice of the datasets favored the ones that sported an accompanying hierarchy on the label set, as these kind of datasets has been suggested Khrulkov et al. (2020) to better match the inductive bias imposed by learning in hyperbolic geometries. We used the Geoopt library Kochurov et al. (2020) to implement the hyperbolic operations. Following the insights provided by Guo et al. (2022), Hamzaoui et al. (2024) and Yue et al. (2024), we apply a clipping of the features with clipping value equal to 1, before projecting them onto the Poincaré ball and we use a temperature $\gamma = 10$.

The experiments were run on a cluster with 4 ARM machine, which consists of Ampere Altra Q80-30 CPU (80-core Arm Neoverse N1), 512GB of memory, 2 x NVIDIA A100 GPU (40GB vram). We used geoopt [34] to implement the hyperbolic operations.

## A.2 HYPERPARAMETERS SELECTION

In our experiments, we tried to replicate the settings in competitor papers when possible. Specifically, the basic configuration of RHPN has been selected to match the training settings from Landrieu & Garnot (2021). We did run a few exploratory experiment to evaluate better settings, but we rapidly found out that the given setting was already a very good one.

The most challenging method to reproduce was HBLGhadimi Atigh et al. (2021). To reproduce it, we used the official repository of the paper. For CUB and Cifar100 datasets, we adopted the settings provided in the original paper as these datasets were already part of the experimentation therein. The only change with respect to this setting, was the adoption of a ResNet18 instead of a ResNet32, which was necessary to ensure a fair comparison with other methods. This change explains why some of the results in our experiments do not match the ones in the original paper. The comparison remain fair since all methods have been tested using the same backbone networks. Also, the main results from the Ghadimi Atigh et al. (2021) paper remain valid: the HBL method continues to outpeform HPS in low dimensional spaces.

Cars and Aircraft were not discussed in the papers that introduced competitor methods. In these cases, for each dataset, we started with the same hyper-parameters we adopted for the dataset that was most similar in terms of number of classes: for Cars we adopted a setting similar to CUB, for Aircraft we adopted a setting similar to Cifar100. We then finetuned the slope parameter for HBL over a separate validation set, resulting in a slope of 0.001 for Cars and of 0.01 for Aircraft.

The slope parameter of RHPN was set testing a few values over a separate validation set. Specifically, we tried values of $\lambda$ in $\{0, 0.01, 0.1, 0.3\}$. Since the similarity in terms of number of classes between CUB and Cars and between Aircrafts and Cifar100, we finetuned $\lambda$ on CUB and Aircrafts and adopted the results over Cars and Cifar100 respectively.

