# OpenReview forum: "Riemannian Optimization for Hyperbolic Prototypical Networks"
_ICLR.cc/2025/Conference — ICLR 2025 Conference Withdrawn Submission_

### Official Review · Reviewer_eihX · 2024-10-23

**Soundness:** 2
**Presentation:** 3
**Contribution:** 1
**Rating:** 3
**Confidence:** 5

**Summary:**

This paper proposes a method for optimizing prototypes in Riemannian manifolds. In addition to the contrastive loss, the authors add a regular term to push the data points away from the boundary. They use the Riemannian SGD method to learn backbone parameters and prototypes. The authors also conduct experiments on image classification. tasks.

**Strengths:**

Strength:
1. Learning in hyperbolic spaces is a promising direction.
2. The experiments have shown the method's effectiveness on the four datasets.

**Weaknesses:**

Weakness:
1. For me, the novelty is limited. Learning prototypes in hyperbolic spaces is not novel, such as [a,b,c,d]. What is the contribution in this paper about prototype learning.

[a] Hyperbolic Image Embeddings
[b] Curvature generation in curved spaces for few-shot learning
[c] Curvature-adaptive meta-learning for fast adaptation to manifold data

2. The experiments are weak. This paper only evaluates the performance of four image classification tasks. Few-shot learning, incremental learning, and zero-shot learning are also used to evaluate the performance of methods in hyperbolic space [e,f,g].
[e] Exploring data geometry for continual learning
[f] Kernel Methods in Hyperbolic Spaces
[g] Clipped Hyperbolic Classifiers Are Super-Hyperbolic Classifiers

3. Since the work focuses on Riemannian optimization, some Riemannian optimization methods should be added to the related work, and the difference between them and this work should be claimed. In addition, the novelty of the Riemannian optimization should be claimed.

**Questions:**

1. The regular term is an important component in this method. However, its motivation and effectiveness are less explored.

---

> ### Author Response · Authors · 2024-11-22
>
> Point 1) We acknowledge that our work offers incremental advancements rather than groundbreaking novelty. However, it is important to emphasize that the proposed architecture surpasses the current state-of-the-art, which constitutes a significant achievement in our opinion. This performance not only highlights the merit of our contribution but also suggests that it is likely to be not trivial—if it were, the state-of-the-art would have already achieved similar or superior results. It is also worth mentioning that the level of novelty is usually subjective in that every new paper builds on older works. We believe that our work is a valuable contribution to the field, as it shows new potential for hyperbolic prototype learning.
>
> Point 2) Thank you for this comment. While we believe that few-shot learning would be a great extension to our work, we currently believe that it poses non-trivial problems, making it difficult to compare with [e]. Specifically, few-shot prototypical networks, usually work by assuming that the prototype for a given class coincides with the centroid of the few examples we had for that class. This is not applicable to our work and, therefore, we prefer to defer it to future work.
>
> Our work focuses mainly on image classification, which is poorly explored by the literature.  We decided to focus on the dimensionality reduction challenge of prototypical networks, and on the regularization needed for the hyperbolic methods to outperform the Euclidean methods. In our opinion, this is also a very interesting aspect of hyperbolic manifolds, but we would like to extend our effort to other scenarios, using the aspects discovered in our experiments.
>
> Point 3) Thank you for pointing this out, now we are starting to believe that perhaps the title was deceiving. We leverage Riemannian optimization in the limited context of prototype learning and are not aware of any work comparable to ours. We will be glad to take into account any relevant paper the reviewer might suggest.
>
> Questions: We sincerely appreciate the reviewer’s emphasis on the importance of the slope parameter. We believe we had adequately explored its impact through Figures 4, 7, and 8, but we would be grateful for any specific metrics or additional analyses the reviewer might suggest to further strengthen our evaluation.

---

### Official Review · Reviewer_SrhE · 2024-10-23

**Soundness:** 2
**Presentation:** 3
**Contribution:** 1
**Rating:** 3
**Confidence:** 4

**Summary:**

This paper performs hyperbolic prototype learning, where the prototypes are made learnable. Moreover, the method adds clipping and regularization to improve performance. The resulting method is compared on four image classification datasets against several other prototype-based methods. They find reasonable improvements over these methods.

**Strengths:**

The structuring of the paper is clear, with an easy to understand presentation of the method and the results. Moreover, the paper contains many visualizations of the training behaviour of the model for various settings.

The comparison between learnable prototypes and fixed prototype methods are interesting to see.

**Weaknesses:**

My main concern is that the paper has limited novelty. While the paper is the first to perform hyperbolic prototype learning with learnable prototypes, this type of setting has been explored before in the usual Euclidean prototype learning setting as mentioned by the authors. Moreover, learnable prototypes can easily be optimized using Riemannian optimization, similar to any other parameter of a hyperbolic model. So, exploring learnable prototypes in hyperbolic learning does not require any technical contributions. The clipping and regularization that the method uses are also coming from other papers, so it seems to me that the paper introduces no new ideas and simply applies existing concepts.

The comparison between the hyperbolic method and the Euclidean method does not seem completely fair to me. RHPN* uses regularization, whereas it seems that ECL does not use any regularization. As such, the fair comparison for assessing the importance of hyperbolic geometry would be between ECL and RHPN. Indeed, when considering this comparison, the hyperbolic solution is a lot less favorable, even being considerably worse on CUB for large embedding dimension. Based on this, I think the choice for hyperbolic geometry is not well motivated, especially considering the increase in computational complexity.

**Questions:**

Hyperbolic geometry is usually beneficial compared to Euclidean geometry when there is a clear hierarchical structure present in the data. Could the method potentially be changed to incorporate hierarchical knowledge in the prototypes that aids in training a classifier for example? That would make the choice for hyperbolic geometry more obvious and, possibly, more impactful.

---

> ### Author Response · Authors · 2024-11-22
>
> We acknowledge that our work offers incremental advancements rather than groundbreaking novelty. However, it is important to emphasize that the proposed architecture surpasses the current state-of-the-art, which constitutes a significant achievement in our opinion. This performance not only highlights the merit of our contribution but also suggests that it is likely to be not trivial—if it were, the state-of-the-art would have already achieved similar or superior results. We believe also that the level of novelty is usually subjective in that every new paper builds on older works. We believe that our work is a valuable contribution to the field, as it shows new potential for hyperbolic prototype learning.
>
> For what concerns the comparison with other methods, ECL does not present any regularization term. The method is exactly the one presented in "Leveraging class hierarchies with metric-guided prototype learning" by Landrieu and Garnot. To augment that method with a regularization term would amount to building a different method, and any shortcomings of the method might then be attributed to the regularization term.
>
> We agree that RHPN alone is not enough to outperform ECL, which would have been a dramatic finding, as the embedding manifold would be the only change. The regularization term is particularly effective in making the embeddings more hyperbolic, shrinking them to the origin. In particular, for the datasets mentioned, the tuning of the slope parameter turns out to be a killing aspect to outperform the Euclidean geometry.
>
> Regarding the comment: "Based on this, I think the choice for hyperbolic geometry is not well motivated, especially considering the increase in computational complexity.", we should stress that the proposed method is only very slightly more computationally complex than the Euclidean counterpart. The differences are the exponential map and the computation of Riemannian gradients, which is not more expensive than doing gradient descent on a network with a similar number of parameters using a loss function that combines two different criteria (as it is common in regularized models). We will make sure to clarify this in the revised version of the paper.
>
> For what concerns the question, we agree with the reviewer. Incorporating hierarchical knowledge can be beneficial in this context, as shown in the existing literature about hierarchical classification, so we thank the reviewer for this suggestion. In our work, we decided to focus on the effectiveness of our method in different dimensional spaces as done in HBL, while treating datasets that have an intrinsic hierarchical structure, but using this hierarchical structure explicitly can be a direction for future works.

---

> > ### Comment · Reviewer_SrhE · 2024-11-25
> >
> > Thank you for your response. I agree that novelty is something subjective and I remain of the opinion that this work has limited novelty. Moreover, I still do not believe that the comparison is fair, even if ECL was originally presented without regularization. Adding a regularization term to ECL and comparing it against RHPN* seems very possible and that experiment would clearly show whether the choice for hyperbolic space makes sense or not. As it stands, the experiments seem to indicate that ECL with regularization might have given better results than RHPN*. Lastly, hyperbolic space is locally similar to Euclidean space, so if you add a term that pushes everything towards the origin, then that is making your representations less hyperbolic, which is another argument against using hyperbolic space.

---

> > > ### Author Response · Authors · 2024-11-26
> > >
> > > Thank you for engaging in the discussion and for sharing your perspective on ECL. We realize there might be a slight misunderstanding regarding the implementation of the ECL algorithm in our work. To clarify, the ECL algorithm is regularized using weight decay, which is applied to all network parameters, including those corresponding to prototypes. For the regularization term, we adhered to the same hyperparameters as specified in the original paper. We hope this resolves any confusion on this point.
> > > We sincerely thank the reviewer for their insightful comment regarding hyperbolicity. However, we are unable to fully grasp the connection you suggest between the local similarity to Euclidean space and the hyperbolicity of points near the origin. In fact, our empirical evidence suggests the opposite to be true, as demonstrated in Figures 5 and 7. Could you kindly clarify your statement further and/or provide references to support this claim? This would greatly help us in addressing your observation more thoroughly.

---

### Official Review · Reviewer_26WM · 2024-11-02

**Soundness:** 2
**Presentation:** 3
**Contribution:** 2
**Rating:** 5
**Confidence:** 3

**Summary:**

This paper introduces Riemannian Optimization for Hyperbolic Prototypical Networks (RHPN), a novel approach that leverages hyperbolic geometry within a Prototype Learning framework.  Specifically, RHPN uses Riemannian optimization on the Poincaré ball to update prototypes during training, enhancing performance with a regularization term. Extensive experiments demonstrate RHPN's superiority over state-of-the-art methods in Prototype Learning across various dimensions.

**Strengths:**

1. The paper introduces the RHPN, a method that leverages the Poincaré ball model of hyperbolic geometry to dynamically update prototypes during training for enhances prototype learning.

2. The paper provides a comprehensive experimental evaluation, demonstrating the effectiveness of RHPN across multiple datasets and embedding dimensions. The analysis of δ-hyperbolicity of the learned embeddings contributes to the understanding of how hyperbolic geometries align with the intrinsic structure of data, offering valuable insights into the geometric properties of the representations.

3. The framework's flexibility to be extended to other geometries is a significant strength, indicating the potential for future research and applications in diverse domains where different hyperbolic geometries might be more suitable.

**Weaknesses:**

1.This work aims to construct a prototype learning framework on generic Riemannian manifolds, with a particular emphasis on embedding prototype representations as network parameters within the Poincaré ball model. However, the paper's main contribution remains somewhat obscured. Although the related works section acknowledges existing studies on prototype learning in hyperbolic manifolds, the paper's unique focus on the application to image classification tasks, as highlighted on lines 049 and 119, is not clearly articulated, which diminishes the perceived novelty of the research.

2.The description of vanishing gradient problem in Riemannian SGD could be more explicit and comprehensive. It would significantly enhance the paper if the authors could provide a clearer illustration of this issue in Figure 3.


3.The discussion on the choice of hyperparameters, particularly the slope parameter , could be more detailed. It is noted that Figure 4(a) and Figure 4(b) illustrate a significant performance variance based on the value of 𝜆, especially when comparing high and low embedding dimensions. Clarifying the reasons behind these performance fluctuations would provide deeper insights into the sensitivity of the model to 𝜆. Furthermore, there seems to be the model collapse during training for 𝜆=0.1 and 𝜆=0.3 as depicted in Figure 4(b) is concerning.


4.The experimental results among datasets indicate an improvement in performance for the image classification task with increasing embedding dimensions. This trend is intriguing and warrants further exploration. Are there any potential bottlenecks or limitations associated with this observation?

**Questions:**

Please refer to the weaknesses.

---

> ### Author Response · Authors · 2024-11-22
>
> Point 1) Thank you for this comment. For the sake of clarity, we recall that, at the end of Section 1, we tried to summarise our main contributions, which are: the introduction of a new framework to update prototypes in Riemannian manifolds; the exploration of key aspects of learning within this framework (mainly, the study of how the lambda parameter term impact on it); an empirical evaluation showing that the proposed technique is able to outperform the current state of the art.
> For what it concerns our focus on image classification: we believed that image classification was a nice application of our technique, one that is less explored in hyperbolic literature. It is worth noting that nothing in our method is specific to image classification, and it can be applied to any other domain. We thank the reviewer for letting us notice that this aspect was not clear enough.
>
> Point 2) Thank you for pointing this out. Learning on hyperbolic manifolds presents a vanishing gradient problem due to the embeddings that naturally collapse to the border of the Poincare ball. However, this is not specifically related to the Riemannian SGD. We wonder if the reviewer agrees with this description. In that case, we can try to be more articulate in the paper.
>
> Point 3) Thank you for this suggestion. We do agree that lambda is a critical parameter in our method. Unfortunately, there is no space, nor time, to delve more into this aspect. In our paper, we made sure to emphasize the importance of this parameter (mainly in the section describing the experiments) and despite it being a very interesting topic, we prefer to leave it for future work.
>
> Point 4) We agree with the reviewer that it is intriguing to study the performance with respect to different embedding dimensions. It is intuitive that there is a bottleneck in learning in low-dimensional spaces, and hyperbolic manifolds are particularly practical in this scenario. In fact, RHPN shows leading performances. However, our findings in this regard are empirical rather than theoretical. Trying to find a theoretical trade-off between dimensionality reduction and performance is an exciting avenue for new research, but not trivial at all and we prefer to leave it for future work.

---

### Official Review · Reviewer_1ESb · 2024-11-05

**Soundness:** 3
**Presentation:** 2
**Contribution:** 3
**Rating:** 6
**Confidence:** 3

**Summary:**

This paper presents Riemannian Hyperbolic Prototypical Networks for Prototype Learning. The method optimizes prototypes on Riemannian manifolds, specifically the Poincaré ball, leveraging hyperbolic geometry. The approach improves Prototype Learning by dynamically updating prototypes during training and incorporating a regularization term for embedding norms. Experimental results show the effectiveness of the proposed method.

**Strengths:**

1. Applying Riemannian optimization in hyperbolic spaces to prototype learning is interesting.

2. Experiments on various datasets demonstrate the effectiveness of the proposed method.

**Weaknesses:**

1. More intuitive explanations for the methodology and mathematical concepts are needed for better accessibility.

2. The Riemannian optimization may add extra computational complexity to the model. which is better to also be investigated and discussed in the paper.

3. The slope parameter $\lambda$ plays a significant role, but its tuning can be complex, as demonstrated by the need for separate tuning across different datasets.

**Questions:**

I'm also curious about whether the method can and how the method performs on few-shot learning, where the prototype-based method is more commonly used.

---

> ### Author Response · Authors · 2024-11-22
>
> Point 1) Thank you for the suggestion, is there any particular definition/concept that the reviewer would like to be better explained? While we did our best to be as clear as possible, an outside perspective is always helpful and will greatly apreciate any suggestion.
>
> Point 2) Thank you for the insights. We did not delve into the computational complexity of Riemannian optimization since, while different, the computational complexity is usually very similar to the Euclidean counterpart. Gradients computation in prototypical networks only requires an additional gradient computation w.r.t. the protypes, which is no more expensive than doing gradient descent on a network with a similar number of parameters using a loss function that combines two different criteria (as it is common in regularized models).
>
> Point 3) We totally agree with the reviewer. Unfortunately, the lambda parameter is very sensitive and, as shown in the experiments, can bring to a huge improvement in the performance. We delve into this aspect with Figures 4 and 8, showing the possible harm and gain that tuning this parameter can bring. In particular, the tuning of lambda enables RHPN to outperform the Euclidean geometry, showing much more robustness to dimensionality reduction of the embedding space. While we believe that a sufficient portion of the paper has already been devoted to this aspect,  we are open to any suggestion on how to better explain it.
>
> Question) We kindly thank the reviewer for this comment. While we believe that few-shot learning would be a great extension to our work, we currently believe that it poses non-trivial problems. Specifically, few-shot prototypical networks, usually work by assuming that the prototype for a given class coincides with the centroid of the few examples we had for that class. This is not applicable to our work and, therefore, we prefer to defer it to future work.

---

> ### Comment · Reviewer_1ESb · 2024-11-25
>
> Thanks for your responses. Most of my concerns have been addressed and I would like to maintain my original rating.

---

### Note · Authors · 2024-12-19

**Comment:**

We thank all the reviewers for their feedback. We will improve the work for future submissions.

**Withdrawal Confirmation:**

I have read and agree with the venue's withdrawal policy on behalf of myself and my co-authors.